# Tracking People with 3D Representations

**Jathushan Rajasegaran, Georgios Pavlakos, Angjoo Kanazawa, Jitendra Malik**
UC Berkeley

## Abstract

We present a novel approach for tracking multiple people in video. Unlike past approaches which employ 2D representations, we focus on using 3D representations of people, located in three-dimensional space. To this end, we develop a method, Human Mesh and Appearance Recovery (HMAR) which in addition to extracting the 3D geometry of the person as a SMPL mesh, also extracts appearance as a texture map on the triangles of the mesh. This serves as a 3D representation for appearance that is robust to viewpoint and pose changes. Given a video clip, we first detect bounding boxes corresponding to people, and for each one, we extract 3D appearance, pose, and location information using HMAR. These embedding vectors are then sent to a transformer, which performs spatio-temporal aggregation of the representations over the duration of the sequence. The similarity of the resulting representations is used to solve for associations that assigns each person to a tracklet. We evaluate our approach on the Posetrack, MuPoTs and AVA datasets. We find that 3D representations are more effective than 2D representations for tracking in these settings, and we obtain state-of-the-art performance. Code and results are available at: `https://brjathu.github.io/T3DP`

## 1 Introduction

Humans are three-dimensional beings, who live and move in a three-dimensional space. However, monocular people tracking algorithms from the computer vision community, e.g., (4; 7; 12; 28; 36; 41; 46) are typically based on 2D representations of people in 2D space. This made a lot of sense in the 1990s when people tracking first came to the fore as (1) the application context was often video surveillance, where people are at low resolutions (2) there were not reliable algorithms for lifting people from 2D to 3D. But today neither of these limitations hold. Applications of video analysis can use HD video (1080p or better), and we have excellent technology e.g., (17) for 3Dfying people.

Figure 1 illustrates why tracking people might be easier in 3D than in 2D. Even when two people overlap and occlude each other in 2D, their spatial location is typically separable in 3D. While 2D appearance features are sensitive to viewpoint and body pose, 3D appearance features are less affected by viewpoint and pose changes. These properties enable robust tracking through people-to-people occlusion and even shot changes, a common phenomena in edited media and online videos.

Tracking people in 3D also opens up many downstream tasks such as predicting 3D human motion from video (18; 21), predicting their behavior (11; 48), and imitating human behavior from video (31). These tasks involve videos of people other than pedestrians, such as videos of people performing sports, dance, daily activities (10), or even edited media such as movies or TV shows (13; 30).

Our input is a monocular video with detected bounding boxes. On each bounding box, we develop an approach that extracts both 3D geometry and appearance information of the person. Specifically, we extend HMR (17), a method that predicts 3D human mesh from monocular images, to also extract the appearance information of the person by predicting a flow field that estimates the texture map, which is a viewpoint and pose invariant space. We refer to this model as Human Mesh and Appearance Recovery (HMAR). From HMAR output, we extract 3D appearance, 3D pose, joints,

3D location is more separable than 2D location

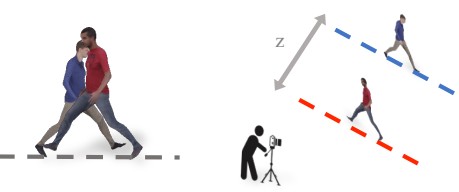

2D representation            3D representation

3D appearance is invariant to viewpoint and pose

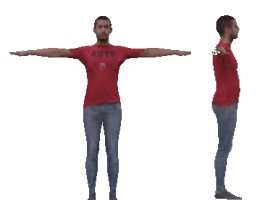 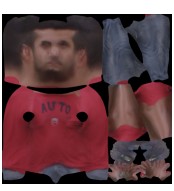

2D representation            3D representation

**Figure 1:** *The benefits of tracking people in 3D.* Left: Using the 3D location of people makes it easier to separate them than using their 2D location or 2D bounding boxes. Right: The appearance of people, when computed in a 3D representation (texture map) is less affected by changes in viewpoint and pose.

and the estimate of their 3D location as separate features. We then aggregate this per-bounding box 3D information across time and space via a transformer (38) that takes these separate 3D cues as input. The transformer acts as a spatio-temporal diffusion mechanism that can propagate information across similar features by means of attention. Finally, we associate the diffused features across time to assign identities to each bounding box, which produces the final tracklets. The overview of our approach is shown in Figure 2.

We evaluate our algorithm on three different datasets: PoseTrack (1), MuPots (27) and AVA (13). AVA is a particularly interesting dataset as it is derived from movies, which include shot changes. These exhibit much greater variety of behavior than videos in the traditional tracking challenges such as MOT. In addition, people are larger (in terms of number of image pixels), which is essential for the 3D approach to work. We find the following: 1. 3D representations are more effective than 2D representations, 2. Ablation studies show that using 3D information in both appearance and location is helpful, 3. We outperform current state of the art on these datasets.

## 2   Related work

**Tracking**    The tracking literature is vast and we refer readers to (7) for a comprehensive summary. Tracking may be designed for generic categories such as any bounding boxes (52), here we focus on methods that track humans. These approaches mostly depend on 2D location or keypoint features (12; 36; 41) and sometimes 2D appearance (4; 28; 43; 46). Most modern-day tracking approaches focus on evaluating their performance on the MOT benchmark (7), which mainly focus on high-density pedestrian scenes observed in outdoor or indoor settings. Under this setting, detection plays a key role and many methods jointly solve for both detection and association (4; 28). In this work, we are interested in the effectiveness of 3D representations for tracking and thus assume that detection bounding boxes are provided, which we associate through our representations. Furthermore, we are interested in tracking people in every day Internet videos where people occupy larger portion of the scene, performing activities such as sports or dancing, and even edited media such as movies.

There are methods that incorporate 3D information in tracking, however these approaches assume multiple input cameras (23; 49) or 3D point cloud observation from lidar data (40). Here we focus on the setting where the input is a monocular video. This setting has also been addressed in the past, e.g., (5), however, here we leverage much more expressive and fine-grained 3D information, including detailed representations for appearance and 3D joint locations, which help to improve the overall tracking performance.

**Monocular 3D human reconstruction**    There is long history of methods that predict 3D human structure or motion from monocular images or videos (6; 14). Recently, there has been rapid progress in this area due to the emergence of statistical models of human bodies such as SMPL (25) that provide a low dimensional parameterization of a deformable 3D mesh of human bodies. Human Mesh Recovery (HMR) method (17) and follow-up works, e.g., (3; 22), employ convolutional neural networks to predict the parameters of an existing body model from a single image. Extending this result in the temporal dimension, HMMR (19) and VIBE (21) reconstruct mesh output for a single person given a series of tracked frames or tracklets as the input. Extensions to movie data have also been investigated in (30). Other works focus on multiple people (16; 37; 47) and regress the

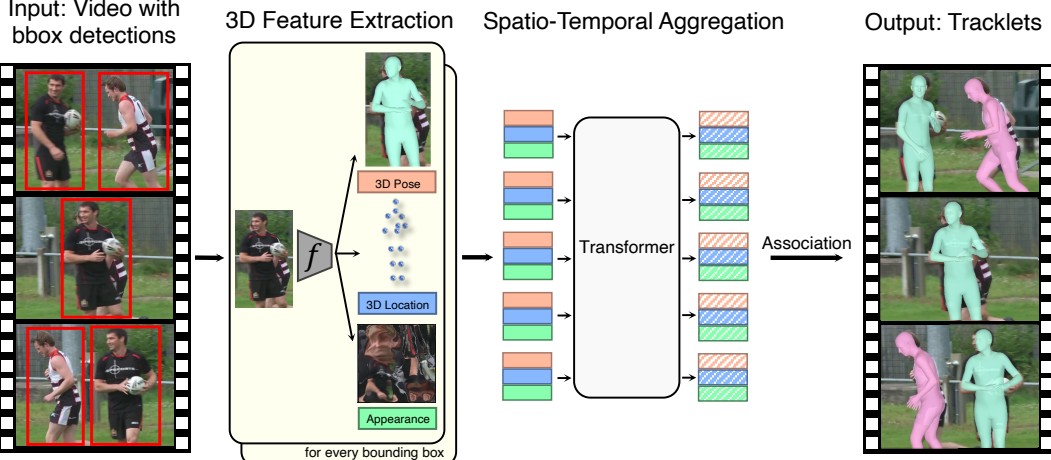

**Figure 2:** *Overview of our framework.* The input to our pipeline consists of a video sequence with bounding box detections. For each bounding box, we use our HMAR model we extract 3D representations corresponding to the person's 3D appearance, pose, and location. The 3D representations for all bounding boxes are then processed jointly by a transformer which aggregates information across space and time. Finally, an online association step assigns identity to each detected human, which constitute the tracklets.

parameters corresponding to multiple people on the same input frame. All of these methods rely on detected bounding boxes or tracklets in the case of temporal domain. As such, these approaches rely heavily on existing tracking algorithms to extract long tracklets associated to the same individual. In this work we explore how the predicted 3D human information may be used to empower tracking in exchange. There are two recent works that jointly solve tracking and 3D pose estimation of multiple people from monocular video (26; 33). While these approaches focus on the task of 3D pose estimation, we focus on the problem of identity association and study the effectiveness of 3D vs 2D representation and compare extensively against state-of-the-art 2D tracking methods.

Related to our approach are recent methods that extract the texture of a person along with their geometry (29; 39; 45). The first recovers the texture only for the visible areas of the body, and uses it to improve the 3D reconstruction, while the other two use identity labels to learn an encoder that predicts complete 3D appearance expressed as UV images. In this work we explore how UV images can be used for associating people across time for tracking.

## 3 Method

In order to assess the effectiveness of 3D information in tracking, we propose a simple tracking framework that assumes as input a monocular video and a set of detected bounding boxes. We then explore how to associate these bounding boxes across time using various cues.

Our tracking algorithm consists of two main modules: our proposed HMAR model, which encodes humans into a rich embedding space, and a transformer model for learning associations between detected humans across multiple frames. The HMAR model learns to predict appearance, pose, and keypoints. These are essential cues for tracking, and each of them carries complementary information. For example, when a person is temporarily occluded, the appearance is important to establish its identity after re-appearance, while when many people share similar clothing in a video, pose and location become the primary cues for tracking. Given this rich embedding of a person, we need to learn associations between different human identities so that each person can be matched in the upcoming frames. We train a modified version of a transformer (38) to learn these associations across multiple human identities.

### 3.1 Human Mesh and Appearance Recovery (HMAR)

Our starting point for 3D feature extraction is the popular HMR architecture (17). Given a bounding box of a person, HMR regresses the 3D pose and shape parameters of the person. While the 3D pose

provides rich geometric information about the person, it throws away the appearance information, which is essential for tracking. To augment this, we extend HMR such that it can also recover the 3D appearance of the person by means of a texture image, which is a space that is viewpoint and pose invariant. We call this method HMAR, which stands for Human Mesh and Appearance Recovery.

In order to recover the texture image, we create skip connections from the spatial features of the Resnet-50 backbone used in HMR and connect them to a new appearance head that maintains the spatial alignment of features through a series of convolution and bilinear upsampling layers (Figure 3). Please see the supplemental for details of the architecture. The appearance head takes these spatial features over multiple scales as input and is trained to predict the texture of the person, using the canonical SMPL texture mapping (25). This is done in the form of appearance or texture flow in the spirit of (18; 51). Specifically, we regress a texture flow $\mathcal{F}$ that has the same dimensions as the texture map. Each value of this flow indicates the coordinates $(x, y)$ of the input image that the corresponding texel is sampled from. Eventually, the final texture map $I_{\text{tex}} = G(I, \mathcal{F})$ is produced by bilinear sampling $G$ of the input image $I$. The predicted texture map is then used to render the textured 3D mesh. The model is trained to minimize the reconstruction loss after rendering the texture and geometry on the foreground pixels using L1 and perceptual loss.

### 3.2   3D Representations for tracking

The tracking performance is greatly affected by the cues and the type of representation used for association. Our embeddings are computed as natural outputs of the HMAR model as follows:

**Appearance:** HMAR recovers the appearance of the person in the form of the texture image $I_{\text{tex}}$. For tracking, we need to create a compact embedding for appearance, so we employ an auto-encoder to further encode this appearance. The auto-encoder takes the texture image as input and is trained to reconstruct it by going through a low dimensional bottleneck. This bottleneck is represented by a 512-dimensional vector $\bar{\mathbf{a}}$, which is used as the representation of the appearance in our pipeline.

**3D Pose:** For the pose representation, we follow previous work (19; 48) and use the embedding $\bar{\mathbf{p}} \in \mathbb{R}^{2048}$ of HMR as a representation of the pose of the person.

**3D Location:** Given an image with dimensions $[W, H]$, HMAR is applied on a square bounding box with center $[c_x, c_y]$ and dimensions $[b, b]$. HMAR also predicts the local camera parameters $[s, t_x, t_y]$ (scale and 2D translation) for the input bounding box. By assuming a constant focal length $f$ for the image, we can recover the approximate 3D global translation of the person, which localizes the mesh in the view coordinate 3D space corresponding to the full image. This translation is expressed as:

$$T = \left[ \begin{array}{ccc} t_x + \frac{2c_x - W}{sb}, & t_y + \frac{2c_y - H}{sb}, & \frac{2f}{sb} \end{array} \right]. \tag{1}$$

Since we do not know the actual focal length, this is only an approximation of the metric location, but we can still recover consistent relative 3D positions of the people over a single video up to scale.

For a more detailed description of the 3D location, we also extract the 3D keypoints of the person that are directly regressed from the recovered mesh, and we position them in the view coordinate 3D space by adding the translation $T$ to them. This representation is strictly more informative than the 2D bounding box information, both because it is in 3D, and because it provides a localized information about the pose of the person than a single bounding box. We also concatenate the temporal information to the 3D keypoints in the spirit of (38), which leads to space-time representation $\bar{\mathbf{s}} \in \mathbb{R}^{90}$.

### 3.3   Spatio-temporal Feature Aggregation

Our HMAR model encodes the 3D appearance, pose, and location information of each bounding box. While this gives us a rich 3D embedding, this representation only captures information from a single-frame. In order to incorporate the spatio-temporal information of the surrounding bounding boxes, we employ a modified transformer model to aggregate global information across space and time. This transformer essentially performs anisotropic diffusion (32) of the representation across all bounding boxes, where the similarity is captured by attention.

Specifically, the input to transformer is the combination of the three different representations obtained from HMAR corresponding to 3D appearance, pose and location, $\mathbf{h} = [\bar{\mathbf{a}}^T, \bar{\mathbf{p}}^T, \bar{\mathbf{s}}^T]^T \in \mathbb{R}^{2650 \times 1}$. The transformer architecture consists of 3 blocks of self-attention and feed-forward layers. We use a

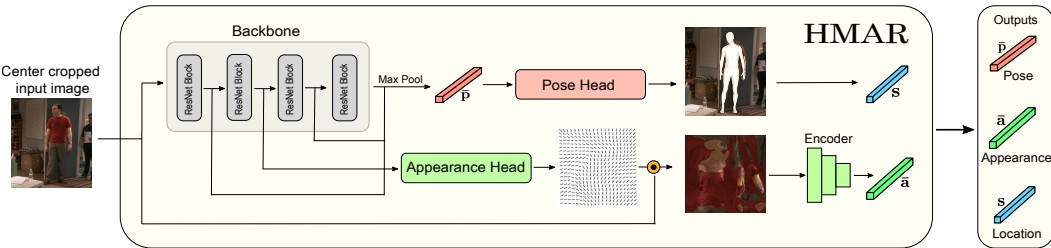

**Figure 3:** *Human Mesh and Appearance Recovery (HMAR):* Our HMAR model learns to predict 3D pose, a UV-image and a set of keypoints relative to the whole image from a single person center cropped image. For the pose, we use the HMR model (17), which gives us a 2048-dimensional pose vector ($\bar{\mathbf{p}}$). We add an additional appearance head to HMR, which learns to predict a flow-field. When the input image is sampled using the flow-field, it produces a UV-image corresponding to the appearance of the person in the image. The UV-image is then encoded into a appearance vector ($\bar{\mathbf{a}}$). Finally, we extract the location of 15 3D keypoints ($\mathbf{s}$) from the 3D mesh. These keypoints represent the position of the human in the 3D space. Therefore, using our HMAR model we can get pose, appearance, and location information of the person.

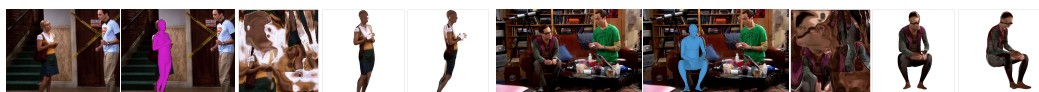

**Figure 4:** *HMAR results.* For each example we show from left to right: (a) Input image, (b) Predicted mesh, (c) predicted texture map, (d) and (e) textured mesh from frontal and side view.

single head attention inside the self-attention layer. During training, we feed $T$ number of frames, each with a maximum number of $P$ people on it. Therefore, for a single clip, the transformer could see $T \times P$ number of humans.

We use a modified transformer that treats each of the three attributes of the $\mathbf{h}^{t,i}$ separately. Specifically, inside the self-attention layer, appearance, pose and keypoints have their own value, query, and key vectors.

$$\mathbf{v}_{app}^{t,i} = W_{app}^v \bar{\mathbf{a}}, \qquad \mathbf{v}_{pose}^{t,i} = W_{pose}^v \bar{\mathbf{p}}, \qquad \mathbf{v}_{loc}^{t,i} = W_{loc}^v \bar{\mathbf{s}} \qquad (2)$$

Here, $W_{app}^v \in \mathbb{R}^{512 \times 512}, W_{pose}^v \in \mathbb{R}^{2048 \times 2048}$ and $W_{loc}^v \in \mathbb{R}^{90 \times 90}$ are the transformation matrices for the value vectors of each attribute in $\mathbf{h}^{t,i}$ vector. Similarly, we can get $[\mathbf{q}_{app}^{t,i}, \mathbf{q}_{pose}^{t,i}, \mathbf{q}_{loc}^{t,i}]$ and $[\mathbf{k}_{app}^{t,i}, \mathbf{k}_{pose}^{t,i}, \mathbf{k}_{loc}^{t,i}]$ vectors by learning projections $W_{att}^q, W_{att}^k$, for $att \in [pose, loc]$. With this setting, we can find attentions for each attribute separately. Finally, the total attention $\mathtt{A} \in \mathbb{R}^{TP \times TP}$ is taken as a weighted sum over various attentions (see also Fig 5):

$$\mathtt{A}^{(t,i),(t',i')} = \sum_{att \in [app, pose, loc]} \beta_{att} \cdot \mathtt{softmax}\left(\frac{\mathbf{q}_{att}^{t,i}(\mathbf{k}_{att}^{t',i'})^T}{\sqrt{dim_{att}}}\right) \qquad (3)$$

Here, $\beta_{app}, \beta_{pose}$ and $\beta_{loc}$ are a set of hyper-parameters. While we use equal values for all of our experiments for simplicity, this design allows for selective attention to different attributes. For example, if we use $[0, 1, 0]$ for $\beta_{app}, \beta_{pose}$ and $\beta_{loc}$ respectively, the appearance and location embeddings are accumulated based only on the attention of the pose embedding. This could be useful for extreme cases, such as when where all people wear the same uniform. Finally, the output of the self-attention layer $\mathbf{h}_{sa}^{t,i}$ is calculated as the weighted sum of the total attention.

$$\mathbf{h}_{sa}^{t,i} = \mathbf{h}^{t,i} + \sum_{t',i'} \mathtt{A}^{(t,i),(t',i')} \cdot [\mathbf{v}_{app}^{t',i'}, \mathbf{v}_{pose}^{t',i'}, \mathbf{v}_{loc}^{t',i'}]^T \qquad (4)$$

We use the same strategy for following feed-forward layers and normalization layers, by treating each attribute separately as in Eq 2. After these human vectors $\mathbf{h}^{t,i}$ have been processed by the transformer, the output vector $\widehat{\mathbf{h}}^{t,i}$ will accumulate information across multiple people over space-time. However, ideally we want $\widehat{\mathbf{h}}^{t,i}$ to attend only the persons with same identities. For example, $\widehat{\mathbf{h}}^{t,i}$ should attend $\widehat{\mathbf{h}}^{t',i'}$ if and only if $i, i'$ are the same person in different frames. If this happens, $\widehat{\mathbf{h}}^{t,i}$ and $\widehat{\mathbf{h}}^{t',i;}$ will

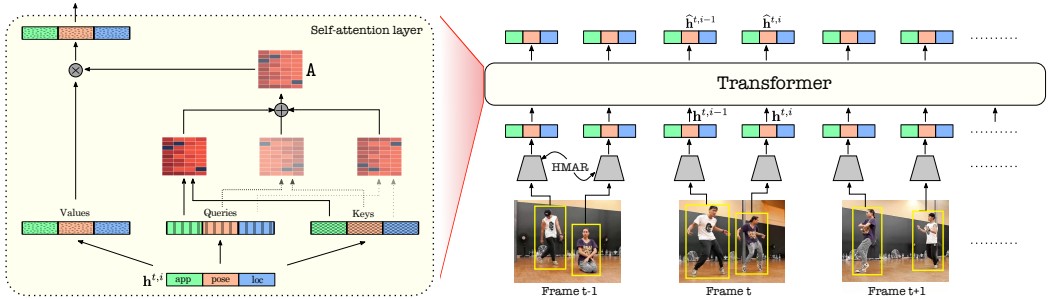

**Figure 5:** *Spatio-Temporal Aggregation with Transformer:* For $T$ frames in a video, we obtain a set of bounding boxes from an off-the-shelf human detector (34). We use HMAR to extract representations of 3D appearance, pose, and location from each bounding box, whose concatenation serves as a token for each bounding box. Since this information is only extracted from a single image, we send the tokens through a transformer, which acts as a diffusion mechanism to propagate information across space and time. Our modified transformer computes attention on each embedding separately, which then gets merged together for a final attention map.

be the same or at least close to each other in some similarity metric $d(\widehat{\mathbf{h}}^{t,i}, \widehat{\mathbf{h}}^{t',i;})$. We enforce this during training by minimizing the ReID (Re-Identification) loss.

$$L_{\text{ReID}} = \sum_{t,t' \sim [1,2,...,T]} d(\widehat{\mathbf{h}}^{t,i}, \widehat{\mathbf{h}}^{t',i'}) + \max\{0, m - d(\widehat{\mathbf{h}}^{t,i}, \widehat{\mathbf{h}}^{t',j'})\} \tag{5}$$

Here, $\widehat{\mathbf{h}}^{t,i}$ and $\widehat{\mathbf{h}}^{t',i'}$ are the embedding vectors for the same person in frame $t, t'$ respectively, while $\widehat{\mathbf{h}}^{t,i}$ and $\widehat{\mathbf{h}}^{t',j'}$ are the embedding vectors for two different people in frames $t, t'$. $\widehat{\mathbf{h}}^{t,i}$ acts as an anchor for the contrastive loss. We use margin $m = 10$ for the negative samples and $L_2$ distance for the distance metric $d$.

### 3.4 Association

Given the 3D representations that have been processed over space-time, the remaining task is to assign identities to each bounding box. We employ a simple association scheme to demonstrate the effectiveness of these representations. Specifically, the output vectors from the transformer $\widehat{\mathbf{h}}^{t,i}$ for all $i \in [P]$ at frame $t + 1$ need to be associated to the existing tracklets up to $t$ or to a new tracklet if no match exists. We use a simple tracking algorithm that takes one frame at a time and assigns track-ids based on the similarities between the past tracklets and the new detections.

First, we start with frame 1, and assign unique identities for each person. This initiates a set of tracklets $\mathcal{T} = \{\mathbf{Tr}_p\}$ for all people $p$ in the first frame. Every tracklet is also associated with the corresponding human embeddings $\widehat{\mathbf{h}}^{1,k}$. Then we move on to frame 2 and calculate the affinity matrix between the detected humans $\mathcal{D} = \{\widehat{\mathbf{h}}^{2,i}\}$ in frame 2, and all active tracklets $\mathcal{T} = \{\mathbf{Tr}_j\}$. We use the Hungarian algorithm to assign detected humans to the tracklets, using the minimum distance between the last 20 embeddings vectors associated with the tracklet and the embeddings vector of the detected human. This process continues for all frames sequentially. In this process, some of the tracklets might be missed and some detections might not be assigned. In some cases, there will be new humans added to the scene, and some people will be occluded or leave the scene. For the first case, we assign a new tracklet to the new person if there are no matches between old tracklets. For the latter case, we keep the old tracklets alive, even if there are no matches for few number of frames. We remove the tracklet if no new person is assigned to the tracklet continuously for 20 frames. A sketch of this tracking algorithm is shown in Algorithm 1.

## 4 Experiments

To evaluate the proposed approach, we first evaluate the efficacy of 3D representations over 2D representations, then provide ablations for the different components of our system and finally compare against state-of-the-art tracking methods. We experiment on three datasets: PoseTrack (1),

---

**Algorithm 1** Tracking Algorithm

---

1: **procedure** ONLINE TRACKING
2: **Require:** $\widehat{\mathbf{h}}^{t,i}$ for all $t \in [T], i \in [P]$, total number of frames $T$, maximum number of people per frame $P$.
3:     $\mathcal{T} \leftarrow \{\mathbf{Tr}_i(\widehat{\mathbf{h}}^{1,i}), \ \ \forall i \in [1, 2, ..., P]\}$ ▷ Initiate tracklets with detections from the first frame.
4:     **for** $t \in [2, 3, ..., T]$ **do**
5:         $\mathcal{D} \leftarrow \{\widehat{\mathbf{h}}^{t,i}, \ \ \forall i \in [1, 2, ..., P]\}$              ▷ Create a detected human set.
6:         Calculate Affinity $\mathbb{A}_{i,j} \leftarrow \mathtt{min}\{\tau, d(D_i, \mathbf{Tr}_j)\}$ for $D_i \in \mathcal{D}$ and $\mathbf{Tr}_j \in \mathcal{T}$
7:         $\mathcal{M}, \mathcal{T}_u, \mathcal{D}_u \leftarrow \mathtt{Assignment}(\mathbb{A})$      ▷ Hungarian algorithm to assign detections to tracklets.
8:         $\mathcal{T} \leftarrow \{\mathbf{Tr}_j(\widehat{\mathbf{h}}^{t,i}), \ \mathbf{Tr}_j(age) = 0, \ \ \forall (i, j) \in \mathcal{M}\}$         ▷ assign detections to tracks.
9:         $\mathcal{T} \leftarrow \{\mathbf{Tr}_j(age) + = 1, \ \ \forall (j) \in \mathcal{T}_u\}$     ▷ Increase the age of unmatched tracks by one.
10:       $\mathcal{T} \leftarrow \{\mathbf{Tr}_j(\widehat{\mathbf{h}}^{t,i}), \ \ \forall (i) \in \mathcal{D}_u, \ j = |\mathcal{T} + 1|\}$     ▷ Add new tracks for new detections.
11:       Kill the tracks with age $\geq 24$.
        **return** Tracklets $\mathcal{T}$

---

MuPoTS (27), and AVA (13), which span a large variety of challenging sequences, including sports, daily activities and storytelling in movies. Since our approach requires mid-to-high resolution images of people, we do not evaluate on the MOTChallenge datasets (7), because of their focus on low resolution humanss. However, we expect that as the technology for 3D human reconstruction is improving, e.g., (44), our approach will be applicable in these settings as well. In all cases, we rely on detections from an off-the-shelf detection system (34), and we solve for the association, assigning identity labels to each bounding box using the proposed approach. As a result, the metrics we employ also focus on identity tracking on the bounding box level. Specifically, we report Identity switches (IDs), or the number of times a recovered trajectory switches from one ground truth tracklet to another. We also report the standard Multi-Object Tracking Accuracy (MOTA) (20) which summarizes errors from false positives, false negatives and ID switches. Finally, we also report the ID F1 score (IDF1) (35) which emphasizes the preservation of the track identity over the entire sequence.

Tracking people in edited media requires robustness against shot changes. For this evaluation we use AVA, which is a dataset of movies with actors annotated with identities. We use the test split from (30), which consists of sequences that contain shot changes. In this setting we *only* evaluate on tracklets that are persistent across shot changes, so to avoid being influenced by detection, we only report IDs and IDF1 on this dataset.

### 4.1 Implementation details

To train HMAR, we use a pretrained HMR model as the starting point. On top of the Resnet-50 backbone that HMR uses, we add additional layers from which we predict the texture flow to predict the texture map. We train this method using images from COCO (24), MPII (2) and Human3.6M (15). We train the appearance head for roughly 500k iterations with a learning rate of 0.0001 and a batch size of 16 images while keeping the pose head frozen. Please see supplemental for more details. After training HMAR, we train an auto-encoder on the predicted texture map, which results in an encoded appearance embedding $\bar{\mathbf{a}}$. We train this using the same datasets as HMAR, and training lasts for about 100k iterations, with a learning rate of 0.0001. Finally, for training the transformer, we use the training set of PoseTrack (1) as this process requires videos with multiple people with labeled IDs for training. Training lasts for 10k iterations with a learning rate of 0.001. All experiments are conducted on a single RTX 2080 Ti.

### 4.2 Quantitative evaluation

#### 4.2.1 3D vs 2D

We first evaluate the effectiveness of 3D representations over 2D representations. To this end, we train a simpler version of our system that only uses one cue and compare with 2D and 3D versions of these cues. The full results of these experiments are presented in Table 1. We experiment with two popular 2D appearance features used in AlphaPose (9; 50) and Tracktor (4), which are convolutional features trained with Re-ID losses. We compare this against the proposed 3D appearance i.e., the encoded texture map. As we can see at the first three rows of Table 1, our 3D-based representation

outperforms the popular 2D-based representations, except on AVA where we perform on par with Tracktor. Regarding the performance of pose/location cues, we compare 3D keypoints with the corresponding 2D keypoints (coming from the projection of the 3D keypoints on the image plane). Again, the 3D-based cue outperform the 2D-based counterpart across the board.

| | Embedding | 2D/3D | Posetrack | | | MuPoTs | | | AVA | |
|---|---|---|---|---|---|---|---|---|---|---|
| | | | IDs↓ | MOTA↑ | IDF1↑ | IDs↓ | MOTA↑ | IDF1↑ | IDs↓ | IDF1↑ |
| App. | Re-ID (OSNet) (50) | 2D | 892 | 63.1 | 77.2 | 88 | 62.7 | 81.2 | 454 | 59.2 |
| | Re-ID (Tracktor) (4) | 2D | 848 | 63.2 | **78.0** | 67 | 62.8 | 79.8 | **242** | **64.6** |
| | Texture map (Ours) | **3D** | **783** | **63.4** | 77.3 | **44** | 62.9 | **82.2** | 246 | 64.2 |
| P&L | 2D pose & location | 2D | 1872 | 59.8 | 74.2 | 194 | 62.2 | **80.8** | 575 | 56.6 |
| | 3D pose & location | **3D** | **1563** | **61.6** | **75.4** | **52** | 62.8 | 80.1 | **280** | **64.2** |

**Table 1:** *Effect of 3D Information:* To assess the effect of 3D information, we train our model using only one embedding that corresponds to either appearance or position/location and compare between the 2D and 3D representations. Using representations based on 3D information performs consistently better than the corresponding 2D-based representations.

#### 4.2.2 Relative importance of different components

Having established the effectiveness of the 3D representation for tracking, we continue with a careful ablation of the components of our system. We report the performance of the pipeline by individually removing each of the 3D cues (i.e., appearance, pose and 3D location). We also remove the spatio-temporal aggregation of the transformer for ablation. The results for the different versions, as well as the full system are presented in Table 2.

These ablations show that 3D appearance is one of the most salient cue across the board. On PoseTrack and MuPoTs 3D location is more important than 3D pose (which is partially subsumed by location since that is based on keypoints). However on AVA, because of shot changes, the two most important cues are appearance and 3D pose, because across a shot boundary 3D pose is preserved while location is not. The transformer consistently adds to the overall robustness of the method. Note that PoseTrack training data is limited (roughly 800 videos each less than a minute), particularly compared to the usual scale of the datasets that transformers are usually trained on, e.g. 300M images for JFT (8). Potentially using larger training data could improve the performance even more.

| Method | PoseTrack | | | MuPoTs | | | AVA | |
|---|---|---|---|---|---|---|---|---|
| | IDs↓ | MOTA↑ | IDF1↑ | IDs↓ | MOTA↑ | IDF1↑ | IDs↓ | IDF1↑ |
| w/o appearance | 1783 | 55.2 | 73.7 | 151 | 64.5 | 78.1 | 739 | 48.7 |
| w/o 3D pose | 711 | 55.6 | 72.2 | 43 | 65.3 | 82.1 | 355 | 65.1 |
| w/o 3D location | 1315 | 54.3 | 69.7 | 59 | 65.3 | 82.8 | 273 | 61.9 |
| w/o transformer | 685 | 49.6 | 64.5 | 41 | 59.1 | 73.7 | 260 | 63.4 |
| Full system | 655 | 55.8 | 73.4 | 38 | 62.1 | 79.1 | 240 | 61.3 |

**Table 2:** *Ablation of the components of our system:* We train our system by removing each cue at a time as well as the transformer for the ablative study. We find that appearance is a salient cue across the board, 3D location is more important than 3D pose in PoseTrack and MuPoTs, but in AVA due to shot changes 3D pose (which is view invariant) is more useful than location. Transformer consistently improves the performance.

#### 4.2.3 Comparison with the state-of-the-art

Lastly, we compare our approach with the state-of-the-art tracking methods. For this comparison, we use four recent popular methods for tracking: Trackformer (28), Tracktor (4), Alphapose (9) and PoseFlow (42). We run these methods out-of-the-box on all benchmarks and report their performance on identity tracking. For a more clear comparison, we provide results with two versions of our approach, one without using the transformer (Ours-v1), and one with the spatio-temporal aggregation of the transformer (Ours-v2). We do this, because our transformer is trained on PoseTrack, but some of the above baselines are not using PoseTrack for training (4; 28), so the Ours-v1 baseline is a more fair choice for a direct comparison. The full results are presented in Table 3. The two versions of our approach perform consistently better than previous work across the board. In Figure 6, we can

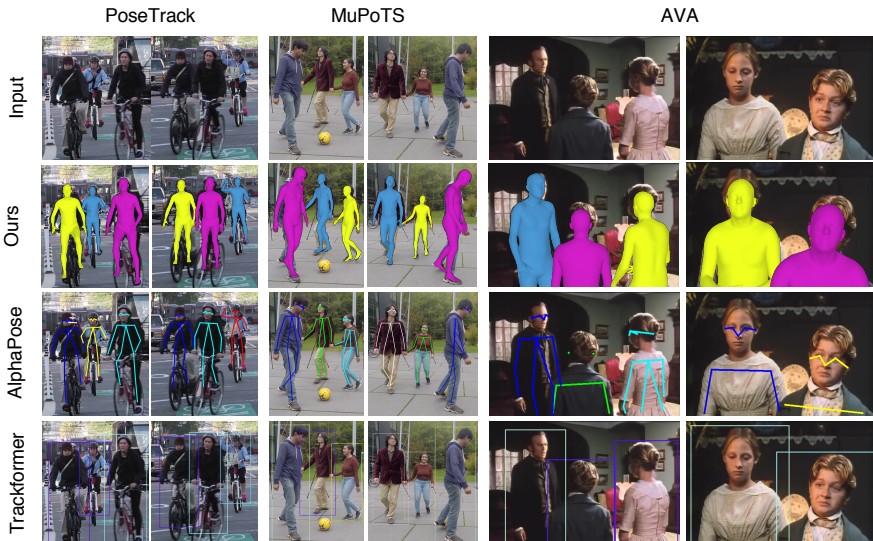

**Figure 6:** *Qualitative results.* We show samples from the three datasets we experiment on, PoseTrack (1), MuPoTs (27) and AVA (13). We provide results for our method, AlphaPose (9) and Trackformer (28). More video results are available in: `https://brjathu.github.io/T3DP`.

also see qualitatively the benefits of our approach compared to these baselines. More specifically, the design choices of our approach make it particularly suitable for challenging cases involving person-person occlusion, or shot changes which are common in edited media (AVA dataset), because our 3D representations improve identity association, when it comes to linking IDs after the person-person occlusions or shot changes.

| Method | Posetrack | | | MuPoTs | | | AVA | |
|---|---|---|---|---|---|---|---|---|
| | IDs↓ | MOTA↑ | IDF1↑ | IDs↓ | MOTA↑ | IDF1↑ | IDs↓ | IDF1↑ |
| Trackformer (28) | 1263 | 33.7 | 64.0 | 43 | 24.9 | 62.7 | 716 | 40.9 |
| Tracktor (4) | 702 | 42.4 | 65.2 | 53 | 51.5 | 70.9 | 289 | 46.8 |
| Ours-v1 | 685 | 49.6 | 64.5 | 41 | 59.1 | 73.7 | 260 | 63.4 |
| AlphaPose (9) | 2220 | 36.9 | 66.9 | 117 | 37.8 | 67.6 | 939 | 41.9 |
| FlowPose (42) | 1047 | 15.4 | 64.2 | 49 | 21.4 | 67.1 | 452 | 52.9 |
| Ours-v2 | 655 | 55.8 | 73.4 | 38 | 62.1 | 79.1 | 240 | 61.3 |

**Table 3:** *Comparison with state-of-the-art tracking methods:* We provide results for two versions of our approach. Ours-v1 does not include the transformer, while Ours-v2 applies the transformer for the spatio-temporal aggregation. Since our transformer is trained on PoseTrack, we provide these two versions for a more fair comparison to methods that do not train on this dataset (i.e., (4; 28)).

## 5   Concluding Remarks

We have demonstrated the value of using 3D representations for tracking people in a wide variety of videos. The proposed approach is simple and modular, and can easily incorporate extra representations such as face embeddings, which may be useful for movies. Our technique relies on having enough resolution on each person to enable lifting to 3D from 2D. Our approach is also susceptible to bad detections, since 3D features computed on non-human regions lead to spurious 3D representations. There are many applications of tracking people in video that have positive societal impact, such as home monitoring for elder care, design of architectural spaces, scientific analysis of child development and more. However, such technology can also be misused for mass-surveillance.

**Acknowledgements:** This work was supported by ONR MURI (N00014-14-1-0671) as well as BAIR and BDD sponsors.

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
