# Supplementary Material for the paper: "Tracking People with 3D Representations"

**Jathushan Rajasegaran, Georgios Pavlakos, Angjoo Kanazawa, Jitendra Malik**
UC Berkeley

## 1 Introduction

In this supplementary document we provide additional information that were not included in the main manuscript due to space constraints. This includes details about a) the method, b) the implementation, c) the evaluation, and d) the visualization. More video results are included in the supplementary video and are also accessible at the anonymous website: `https://sites.google.com/view/t3po/`.

## 2 Method details

In this section, we provide additional details that clarify some components of our proposed approach.

**3D Location approximation:** Originally, the camera prediction is done in the bounding box space, assuming a weak perspective camera, as done by HMR (9). When doing the conversion to the view coordinate 3D space corresponding to the full image, we assume a constant focal length $f$ and a fully perspective camera model. To get a reasonable approximation for the translation $T$ of the person with respect to the camera, we consider the root joint (i.e., pelvis) of the person. More specifically, when projected under the weak perspective camera of the (local) bounding box, the root joint projects to the pixel $(x_i, y_i)$ of the bounding box. We now require that, when projected under the fully perspective camera of the full image, the root joint projects to the corresponding pixel of the full image, i.e., $(x_i + x_c - b/2, y_i + y_c - b/2)$ Solving this set of equation gives us the approximate location $T$ indicated in the main manuscript.

**3D Location encoding:** Regarding the location encoding, originally, we have 15 3D keypoints $\mathbf{s} \in \mathbb{R}^{15 \times 3}$ in the view coordinate 3D space. The vectorized version of $\mathbf{s}$ is encoded into a vector $\mathbf{s}_{loc} \in \mathbb{R}^{45}$ by passing through an MLP. Additionally, we include a temporal token $\mathbf{T}_{time} \in \mathbb{R}^{45}$ to indicate the index of the current frame on the temporal axis. We use sine and cosine function with different frequencies (17) to encode the time index of the frame. Eventually, the space-time embedding $\bar{\mathbf{s}}$ is taken as a concatenated vector of both space and temporal embeddings:

$$\bar{\mathbf{s}} = [\mathbf{s}_{loc}^T | \mathbf{T}_{time}^T]^T \quad \text{where,} \quad \mathbf{s}_{loc} = \phi_{loc}(\mathbf{s}), \quad \mathbf{T}_{time,2i} = \sin \frac{time}{10000^{2i/45}}, \mathbf{T}_{time,2i+1} = \cos \frac{time}{10000^{2i/45}}. \tag{1}$$

Here, $\bar{\mathbf{s}} \in \mathbb{R}^{90}$ is our final space-time representation and $\phi_{loc}$ and $\phi_{time}$ are small two-layer MLP.

## 3 Implementation details

We train our HMAR and our Transformer models separately. First, we train the HMAR model using L1 and perceptual loss (20), weighted equally. Given a bounding box, we feed it to the backbone of the HMR model (ResNet 50). This gives a final layer output (a max pooled vector of 2048, $\bar{\mathbf{p}} \in \mathbb{R}^{1 \times 2048}$) as well as intermediate feature maps from each of the ResNet blocks. The max-pooled vector is then passed to a SMPL-head which regress the camera and the SMPL parameters. Note that, when training the HMAR we are using a pre-trained HMR model, therefore we are not optimizing for the pose during this training. So, the backbone weights and the SMPL-head weights are frozen. We train an appearance-head, which predict a flow-map using the max-pooled vector and the intermediate representations.

35th Conference on Neural Information Processing Systems (NeurIPS 2021).

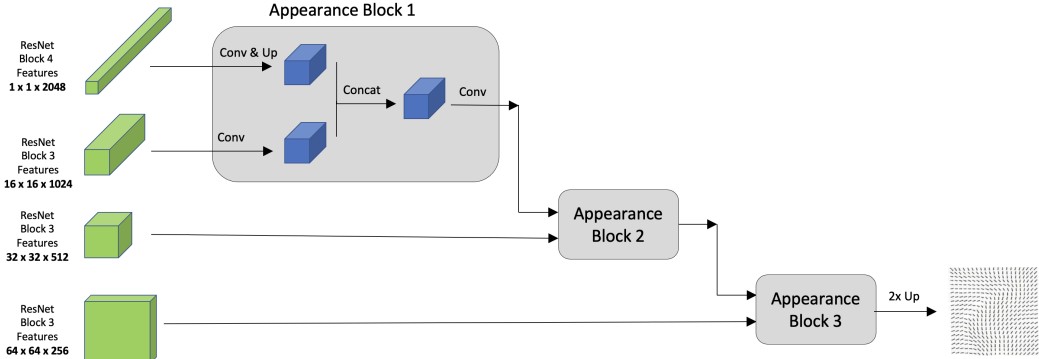

**Figure 1:** *Architecture of our appearance head*

Our appearance head architecture is shown in Fig 1. We take feature maps from all the intermediate blocks of the backbone ResNet of the HMR model. A single appearance block takes two of the feature maps from the ResNet model. For the feature map with smallest resolution, we apply convolution and bi-linear up-sampling, while for the other feature map we only apply convolution. Then, both of these feature maps are concatenated along the channel axis and passed through another convolution. This block is repeated three times. The final flow field is generated by simply up-sampling the feature-maps from the last appearance block.

The predicted flow-map has the same size with the input image, and it is used to sample the pixel values from it. The resulting texture map is then wrapped around the 3D mesh (3D mesh is taken from SMPL model) and re-projected into the 2D image. The L1 loss is calculated between the input image and the projected 3D mesh with texture, while the perceptual loss is computed for the re-projected image. Both losses are weighted equally to compute the total loss. We train the appearance head using images from COCO (13), MPII (2) and Human3.6M (8), for roughly 500k iterations with Adam optimizer, a learning rate of 0.0001 and a batch size of 16 images.

Once the appearance head is trained, we then train a simple auto-encoder which takes a texture map of the input image, and outputs a reconstructed version of it. Between the encoder and the decoder, there is no skip connection, therefore, the intermediate vector (appearance vector) should capture the low-frequency texture information. We train this autoencoder with a simple reconstruction loss between the input and the output. We use the same datasets as HMAR, and training lasts for about 100k iterations, with a learning rate of 0.0001. The encoder has five convolutional layers and five max-pooling layers. This converts $256 \times 256$ resolution UV-map to a $8 \times 8 \times 8$ tensor. We flatten this tensor and use it as the appearance vector ($\bar{\mathbf{a}} \in \mathbb{R}^{1 \times 512}$).

Finally, we train a 3-layer Transformer model, with each layer having single head attentions. Since, the input vectors for the transformer have a mixture of appearance, pose and location information, we compute self-attention for each of the attributes separately. The final attention is taken as a weighted sum of the attentions from pose, appearance and location. We use the training set of PoseTrack (1) as this process requires videos with multiple people with labeled IDs for training. Training lasts for 10k iterations with a learning rate of 0.001. All experiments are conducted on a single RTX 2080 Ti.

Regarding execution, most of the actual runtime is attributed to the initial detection and lifting networks, while the transformer and the simple association scheme are quite fast. We give some indicative runtimes on our system (RTX 2080 single-GPU), but some of these components can be further optimized.

- Detection: Depends on the actual detection system used. For our experimental evaluation, we run offline Faster RCNN, but for a real-time application, one can use a YOLO model, taking ~30ms for a single frame.
- HMAR encoding: Depends on the number of detections, which can also be processed in batch mode. Generally, a forward pass costs 16ms for a single bbox, or ~35ms for 10 boxes.
- Transformer: 8ms for a single forward pass.
- Association Scheme: 2ms per frame.

# 4 Evaluation

**Metrics:** Our primary goal is to solve for the association, given a specific set of detected bounding boxes from an off-the-shelf algorithm, and the metrics we use to report results also reflect this. Specifically, we focus primarily on metrics related to the identity assigned to each detection, and less on detection-related metrics, which also

evaluate the quality of the bounding box detections. We refer the interested reader to (5) for a discussion on metrics and their definition.

**PoseTrack evaluation**: For PoseTrack (1), we report results on the PoseTrack2018 validation set. Typically, the benchmark is used for 2D joint tracking, so most papers report results on the level of 2D joints. However, our focus is on identity tracking, so we compute and report results for the different methods on the person level. This allows us to directly compare methods from the literature that track 2D human bounding boxes (e.g., (3; 15)), and methods that track 2D human skeletons (e.g., (6; 18)).

**MuPoTS evaluation**: For MuPoTS (14), we report results on the 20 test sequences. We use the identity annotations provided with the dataset, and report the same metrics as with the other benchmarks.

**AVA evaluation** For AVA (7), our evaluation focuses on sequences with shot changes. We use the bounding box and ID annotations provided by the dataset and we evaluate on the sequences with shot changes where (at least) one of the identities is visible both before and after the shot change, similar to (16). Thus, the IDs and IDF1 metrics are computed using the bounding boxes of these identities that persist even after the shot change. Since ground truth bounding boxes for AVA are sparse (provided at 1fps) and we only consider these persisting identities, we do not report MOTA, since this metric is dominated by the FP (False Positives) on the non-annotated people. Finally, we clarify that each method runs on the full framerate video, as is typical, and the sparse, 1fps, bounding boxes are considered to report results.

## 5 Additional quantitative results

Here, we investigate two additional design choices for our system. Regarding the 3D pose representation, we use the raw feature vector output from the ResNet backbone, which is also common practice from previous work (10; 12; 16; 19). A reasonable alternative here is to use the SMPL parameters, however, this requires an additional encoding before sending them to the transformer. We evaluate a system, with SMPL parameters as the pose representation. This system performs similar to ours-v1, with a small increase ($\sim 1\%$) on ID switches.

Regarding the tracking system, although the basic version without the transformer (Ours-v1 in Table 3) is purely online, the full system with the transformer (Ours-v2) does consider future frames. However, one can easily run an online version, by controlling the attention masks that are given as input to the transformer - i.e., one can block the embeddings of time t from attending the embeddings of time t+1, thus leading to a causal system. This version performs comparably to the full, non-causal system ( 2% increase in the ID switches).

Finally, for a fair comparison, we also report results on a small subset of videos for each dataset, where this subset is decided by the success of all the methods on tracking these videos. For example, Tracktor fails on about 15 videos of posetrack, therefore we evaluate all other methods on only the rest of the videos. We present these results in Table 1.

| Method | Posetrack | | | MuPoTs | | | AVA | |
| --- | --- | --- | --- | --- | --- | --- | --- | --- |
| | IDs↓ | MOTA↑ | IDF1↑ | IDs↓ | MOTA↑ | IDF1↑ | IDs↓ | IDF1↑ |
| Trackformer (15) | 1143 | 36.4 | 64.5 | 43 | 24.9 | 62.7 | 497 | 43.5 |
| Tracktor (3) | 624 | 42.4 | 65.2 | 53 | 51.5 | 70.9 | 283 | 47.0 |
| Ours-v1 | 596 | 50.6 | 65.7 | 41 | 59.1 | 73.7 | 168 | 62.6 |
| AlphaPose (6) | 1964 | 37.7 | 67.1 | 117 | 37.8 | 67.6 | 631 | 42.2 |
| FlowPose (18) | 923 | 16.2 | 64.6 | 49 | 21.4 | 67.1 | 321 | 52.3 |
| Ours-v2 | 554 | 57.8 | 74.7 | 38 | 62.1 | 79.1 | 169 | 61.1 |

**Table 1:** *Comparison with state-of-the-art tracking methods:*

## 6 Visualization

For the visualization of our results, we use the meshes from the single frame reconstruction. Any inaccuracies to the reconstructed human are not related to our method, but to failure of the single frame HMR model (9) to accurately reconstruct the human body shape. Our method is only producing the association (color of the mesh), but we use this more descriptive mesh visualization, since we believe it makes visual inspection easier. Related to that, we also use a color palette from various studies (4; 11) that makes the tracklet colors more distinctive, and as easy to distinguish as possible For all other methods, we adopt the original visualization used by the authors.