# OpenReview forum: "Tracking People with 3D Representations"
_NeurIPS.cc/2021/Conference — NeurIPS 2021 Poster_

### Official Review · Reviewer_VGK2 · 2021-07-15

**Rating:** 7
**Confidence:** 4

**Summary:**

The submission presents a tracking approach based on 3D representations (i.e., 3D appearance, pose, and location) for tracking multiple people in the video. The paper claims that 3D representations are more effective than 2D representations for tracking humans, which is also validated in several datasets including Posetrack, MuPoTs, and AVA.

**Limitations And Societal Impact:**

The proposed approach can better handle the shot changes in videos, but the results are not so good under the occlusions and overlap between multiple people (see videos in the supplementary material). The statement "particularly suitable for challenging cases involving person-person occlusion" in L274 seems somewhat over-claimed.


**Main Review:**

The paper proposed an approach using 3D Representations for Tracking, which is based on a pipeline for human mesh and appearance recovery and spatio-temporal feature aggregation. Though the recovery of mesh and appearance is not new in literature, jointly using 3D appearance and pose for tracking is somewhat interesting.

+ Overall, this manuscript is well-written and includes technical contributions for tracking multiple people in 3D space.

+ State-of-the-art results in quantitive and qualitative aspects.

There are a few concerns regarding this paper:

- In Sec. 3.2, the global feature outputted from the backbone network is used as the pose representation, how about using the shape and pose parameters of the SMPL model? Though it also makes sense to use deep features as the representation for end-to-end learning, using the SMPL parameters as the pose representation should be also an alternative solution.

- The paper introduces a model for human mesh and appearance recovery, however, there are no quantitive results on the performance of mesh or appearance recovery. It seems that the spatio-temporal feature aggregation can also improve the mesh reconstruction in the video. Adding quantitive results should further strengthen the paper.

- There are also existing approaches related to appearance recovery and reconstruction under the shot changes in videos. It is recommended to including discussions with these papers.

TexturePose: Supervising Human Mesh Estimation with Texture Consistency

Human Mesh Recovery from Multiple Shots

**Time Spent Reviewing:**

6

---

> ### Author Response · Authors · 2021-08-10
> **Reply to Reviewer4**
>
> We thank the reviewer for the valuable comments and time.
>
> **Using SMPL pose and shape parameters for tracking:**
> Using SMPL parameters for the 3D pose representation is indeed a reasonable alternative. However, this requires an additional encoding before sending them to the transformer, so we kept it simple and used the raw feature vector output from the ResNet backbone, which is also common practice from previous work [18,20,28,45]. Since this alternative of using the SMPL parameters was suggested by the reviewer, we experimented with it, and we observed a small decrease in performance for the full system (~5-7% increase in the ID switches).
>
> **Mesh recovery performance:**
> We would like to clarify that the focus of this work is on identity association and not on human mesh recovery. We only introduce HMAR in order to make use of viewpoint and pose-invariant appearance features, which indeed improves tracking performance (Table 2). Regarding temporal pose estimation, the overall system is trained with the Re-ID loss, so some mild adaptations are needed to be able to evaluate the output of the spatiotemporal processing (specifically, training additional outputs and using reconstruction losses to enforce that one can read smoothed pose, appearance, etc).
>
> **Related work:**
> We will extend the discussion for the suggested papers. Briefly, the appearance estimation part of the first paper (“TexturePose: Supervising Human Mesh Estimation with Texture Consistency”) is relevant, but they only recover the texture for the visible areas of the body, and use it to improve the 3D reconstruction, while we leverage appearance as a cue for association estimation. Also, the multi-shot aspect of the second paper (“Human Mesh Recovery from Multiple Shots”) is relevant, but they focus on pose estimation instead, so our approach could be a natural preprocessing step at recovering the tracklets used as input for running their reconstruction.
>
> **Limitations:**
> Since we visualize the reconstruction results from single-frame HMAR, there are indeed limitations when it comes to the reconstruction of people under heavy occlusion. Our main focus is to improve the identity association and we indeed observe improvements compared to previous approaches when it comes to linking IDs after person-person occlusions (e.g., 1:51-2:45 in the supplementary video). We will clarify the statement in L274 to reflect this.

---

> > ### Comment · Reviewer_VGK2 · 2021-08-24
> > **Comment on the rebuttal**
> >
> > Thanks! The rebuttal has addressed most of my previous concerns. I will keep my rating as acceptance. Releasing code of the proposed method could also make this work more impactful.

---

### Official Review · Reviewer_tVEj · 2021-07-16

**Rating:** 6
**Confidence:** 5

**Summary:**

This work introduces a method for monocular multi-person full-body tracking. The method itself is an extension of an existing approach HMR that predicts body meshes from monocular images.
The key idea of this work is to rely on a view- independent appearance representation based on UV texture mapping: one of the model's branches predict a 2D field that is used to sample input image to produce the texture, and is trained to reconstruct the input images. A transformer network is then used to propagate information temporally among similar object hypothesis, and the resulting features are then associated via hungarian matching. Experimental evaluation is conducted on multiple real-world datasets.

**Limitations And Societal Impact:**

Authors do discuss some of the relevant limitations and societal impact.
One issue that might be worth mentioning is the scalability of the attention module for large number of identities.

**Main Review:**


### Writing quality / clarity
**+** The paper is generally clear and well-written, apart from some minor issues (see Misc below).

### Novelty / significance

**+** The general idea of using a UV texture as a view-independent way to represent human appearance makes a lot of sense. It is not particularly novel, but it seems like this work is one of the first ones to exploit it for multi-person tracking.

**-** (minor) It is not strictly correct to say that these UV images are pose-invariant though, as they certainly contain a lot of pose-dependent information (even if they were complete).

**+/-** Using transformer-like architectures for multi-object tracking to enable learnable associations is also a reasonable design choice, albeit not very novel (TransTrack, Trackformer).

**-** One concern about the method is that it assumes that detections are provided as input, and does not really provide a way to improve those based on resulting tracking. One of the reasons to do tracking in the first place is to be more robust to the quality of the detections. This seems like a major limitation, and it would be great to hear authors' comments on this.


### Evaluation

**+** Experimental evaluation indicates that the method outperforms several recent baselines in terms of MOTA and other metrics.

**+/-** Authors provide extensive visual results on in-the-wild sequences, and tracking seems to robust compared to the presented baselines. It does appear that the method is not particularly robust to occlusions, which might be related to the heavy reliance of the method on output of the detections.

**+** Ablation studies are quite thorough, and do indicate that most of the proposed components are necessary to achieve good performance.

**-** It seems strange that authors do not provide comparison to XNect, considering that the model is publicly available. While it is indeed the case that that method provides an alternative way to track individuals (with part association), it does seem quite robust, and still can be used for multi-human tracking.

**+/-** When looking at the resulting tracking videos (and in particular failure modes), it seems a bit strange that for people with significantly different appearance (and body sizes) there are identity switches (e.g. around 03:30 of the video). Would be great to have an intuition on why this happens.

**-** There are no numbers on running time / computational complexity analysis, which is important for the given application and that some of the competing approaches are real-time.

### Misc
* 2048 seems like an incredibly large number to represent a human pose. SMPL has ~80 parameters, what is the reasoning behind using more than an order of magnitude more?
* L4: this serves
* L12: that assign
* L24-L25: * There has been *a lot* of work on tracking people in 3D (more often in multi-view settings). Might make sense to take this into account.
* It might be helpful to visualize sample UV textures to get a feeling of how representative is the appearance information.
* Does this method also work in online settings? Judging from Figure 5, the algorithm is also taking into account "future" predictions.

**Time Spent Reviewing:**

2.5

---

> ### Author Response · Authors · 2021-08-10
> **Reply to Reviewer3**
>
> We thank the reviewer for the valuable comments and time.
>
> **Static detections:**
> In our setting, we assume that we are given a set of bounding box detections, and our method solves for the association. This falls under the popular tracking-by-detection paradigm [6]. Since our motivation is to study the use of 3D representations for the tracking problem, we focus on evaluating and establishing their effectiveness in the association step, without modifications to the input bounding boxes. Clearly, modifying the input detections is important ([6] has an interesting discussion), however, we elected to focus on the data association aspect in order to validate our claims about 3D representations and provide apples-to-apples comparisons. With that being said, our overall pipeline does not fundamentally hinders the refinement of the input detections - for example, one could train additional outputs after the transformer processing that make predictions about 3D poses/locations (and correspondingly 2D bounding boxes) in frames where detections are missing.
>
> **Robustness to occlusions:**
> For our body shape visualization, we use the output of the single-frame HMAR, which means there are some limitations regarding the reconstruction of people under heavy occlusion. However, when it comes to identity association, which is the focus of this work, we do indeed observe clear improvements compared to previous approaches on the aspect of correctly linking identities after the person-person occlusion. The supplementary video (specifically during 1:51-2:45) provides a variety of examples like that.
>
> **Failure cases:**
> Regarding some of the failure cases in our supplementary video (particularly around 3:30), we attribute them to the very close interactions/touching between people, e.g., hugging. Although we observed increased robustness compared to previous systems when it comes to ID association in cases of vanilla occlusions (i.e., a person walking behind another), we believe these close interactions remain challenging for two reasons - a) the people are very close in 3D, which leads to similar 3D locations, and b) appearance estimation becomes more prone to errors, in the sense that the appearances of the two people get mixed (i.e., the appearance of person A can contaminate the appearance features of person B and vice versa), making association harder to establish. We will further discuss this in Section 5 of the paper.
>
> **Running time:**
> Most of the actual runtime is attributed to the initial detection and lifting networks, while the transformer and the simple association scheme are quite fast. We give some indicative runtimes on our system (RTX 2080 single-GPU), but some of these components can be further optimized.
> - Detection: Depends on the actual detection system used. For our experimental evaluation, we run offline Faster RCNN, but for a real-time application, one can use a YOLO model, taking ~30ms for a single frame.
> - HMAR encoding: Depends on the number of detections, which can also be processed in batch mode. Generally, a forward pass costs 16ms for a single bbox, or ~35ms for 10 boxes.
> - Transformer: 8ms for a single forward pass.
> - Association Scheme: 2ms per frame.
>
> We will include this analysis in the final version.
>
> **Invariance for UV images:**
> Our goal when using the terms “view & pose invariant” is to establish that, while the values of 2D pixels in a bounding box can vary wildly depending on the pose of the person, when reasoning in 3D, we can always map the appearance of the person to a “pose-normalized” space (i.e., a specific 3D location on the human body always corresponds to the same location on the texture map). The exact appearance value on a texel can still change sometimes over the course of the video (e.g., different lighting/shadows due to different poses), so we will clarify this in the text to avoid any misunderstandings.
>
> **Scalability for a large number of identities:**
> This is an interesting topic of discussion. The datasets we use for evaluation contain examples with 2 identities (e.g., AVA), all the way up to ~50 identities (e.g., for some PoseTrack videos). When the number of identities increases significantly, some of the errors we observe can be attributed to the lower resolution of each person which can compromise 2D-to-3D lifting (L279-L280) but the attention module seems to scale reasonably well. Still, we believe that with access to more data, the transformer could further improve (L258-L261).
>
> **Response to Misc comments:**
>
> **Size of pose representation:**
> It is common practice among previous works (e.g., [18,20,28,45]) that rely on an HMR-type model to use the features of the ResNet backbone as a neural representation of human pose. We follow the same design choice. One could add more intermediate fully connected layers to bring down the dimensionality before passing the features to the transformer, but this only adds more processing, whereas using the direct output of the convolutional backbone is a convenient design choice. Regarding the use of raw SMPL parameters for the 3D pose representation, please also see the relevant response to R4 below.
>
> **Multi-view works on 3D tracking:**
> Since our focus is on monocular input, we only briefly touch upon multi-view approaches for 3D tracking (L64-L66). We will extend the discussion to better cover this aspect too.
>
> **UV texture visualization:**
> We provide some example visualizations of the estimated UV map and the corresponding textured mesh in Figure 4. The result is not perfect, but the mapping includes the majority of the relevant appearance information that is necessary to solve for association.
>
> **Online version:**
> We clarify that the basic version without the transformer (Ours-v1 in Table 3) is purely online. The full system with the transformer does consider future frames, but one can easily run an online version, by controlling the attention masks that are given as input to the transformer - i.e., one can block the embeddings of time t from attending the embeddings of time t+1, thus leading to a causal system. This performs comparably to the full, non-causal system (~3% increase in the ID switches).

---

> > ### Comment · Reviewer_tVEj · 2021-08-23
> > **thanks!**
> >
> > Thanks for the rebuttal! My concerns have been addressed, and I am keeping my original rating.

---

### Official Review · Reviewer_4pGJ · 2021-07-16

**Rating:** 4
**Confidence:** 5

**Summary:**

This paper presents an approach for multi-people tracking in videos using 3D body representations based on the SMPL model and their associated texture maps. Given an input video, the 2D bounding boxes are assumed to be given and the paper focuses on estimating the tracklets of each person, i.e. solving the temporal association of the bounding boxes.
For this purpose, a first module building upon HMR [Kanazawa 2018] is used to extract the SMPL body parameters for each bounding box. Additionally, the 3D location and appearance of the bounding box is also regressed, the latter using UV texture images.
Then, temporal aggregation of features is performed using a Transformer model. Finally, the transformed features per bounding box are associated to existing tracklets using the Hungarian algorithm.
The main contribution and novelty of this approach is in injecting 3D body representations into the tracking pipeline, jointly with the UV map appearance.
The approach is evaluated on three datasets, Posetrack, MuPoTs and AVA, and demonstrates improved performance w.r.t. state-of-the-art of 2D tracking methods.


**Ethical Concerns:**

The authors also briefly discuss ethical issues in the “Concluding Remarks” section.


**Limitations And Societal Impact:**

The authors shortly mention some limitations and societal impact in the “Concluding Remarks” section.


**Main Review:**

********ORIGINALITY********

As mentioned above, the main contribution of this work is in encoding person information using its 3D body representation and the associated appearance UV map. This allows reasoning about the 3D spatial position of the people, which helps to solve situations with occlusions.

While introducing 3D reasoning into the tracking pipeline is a very promising research line and this improves the most recent state-of-the-art on 2D multi-people tracking (Trackformer [27] and Tracktor [4]) the paper has a number of limitations:

1) The method relies on images with a relatively high resolution of the people. Otherwise, the 2D-to-3D regression would be very unreliable, affecting the overall performance of the approach. In contrast, Trackformer and Tracktor can handle situations with smaller resolution of the bounding boxes.

The evaluation section is not convincing. Concretely:

2) The proposed approach is evaluated on three datasets, Posetrack, MuPots and AVA, and only compared to Trackfomer, Tracktor, Alphapose and PoseFlow. However, many other approaches perform better on these datasets. For instance, in the Posetrack dataset, the work of Wang et al. 2020 “Combining detection and tracking for human pose estimation in videos” reports a MOTA score > 64, both in PoseTrack2017 and PoseTrack2018 test sets. This is significantly higher than the MOTA scores obtained by the submitted approach and Trackformer/Tracktor.  Keytrack [Snower 2019] and FlowTrack [Xiao 2018] also report better results on Posetrack than this submission.

3) MuPots is intended for 3D tracking and standard metrics used to evaluate methods on this dataset are PCK and MPJPE. In order to position the current submission within the state-of-the-art, the authors should compute these metrics and compare with the most recent works. The best results on MuPots are reported by [Cheng et al. 2021] “Graph and Temporal Convolutional Networks for 3D Multi-person Pose Estimation in Monocular Videos”.  A direct comparision with this approach should be done.

In summary, without clearly positioning the current submission within SOTA the reviewer does not believe this paper can be accepted. The proposed methodology is interesting, but the authors focus their comparison on Trackformer and Tracktor, which are designed to track people in more crowded scenes, like those of the MOT17 and MOT20 datasets.


********CLARITY********
The paper in general reads very well, although there are details of the method that remain unclear. For instance:

1) What is the maximum number of people that can be tracked. How does  this this number compared to Trackformer and Tracktor?

2) The regression of the UV maps does not seem correct. As shown in Figure 1, a UV map should contain black regions, as the flattening of the body is not continuous. Also, the different parts of the body (head, torso, legs ...)  should be clearly visible. However, the regressed UV maps (Fig.2 or Fig.3) show a continuous pattern in which the different body parts cannot be distinguished.

********SIGNIFICANCE********

While the approach shows improvement w.r.t. Trackformer and Tracktor in the chosen datasets, the fact that other recent papers have not been considered makes it difficult to really assess the significance of the proposed method.


**Time Spent Reviewing:**

4 hours

---

> ### Author Response · Authors · 2021-08-10
> **Reply to Reviewer2**
>
> We thank the reviewer for the valuable comments and time.
>
> **Resolution of input images:**
> Our method indeed will be more reliable in cases where lifting people to 3D is more accurate, i.e., where the resolution is higher (as a rule of thumb, we consider this to be those where the bounding box for each person is roughly 100 pixels or more). We discuss this aspect in the first paragraph of our intro, L19-L23 and also in the limitations L279-L280. The goal of this work is to explore how to incorporate 3D information in tracking people effectively, and our core observation will carry through as 3D pose reconstruction algorithms improve over time on low-resolution people (a good example is “3D Human Shape and Pose from a Single Low-Resolution Image with Self-Supervised Learning, Xu et al, ECCV 2020). Furthermore, for many video analysis applications beyond surveillance videos, e.g., sports or movies, we have access to high resolution videos. These are important scenarios which are well addressed by our approach (based on the results on PoseTrack and AVA).
>
> **Evaluation: - Comparisons - MOTA metrics:**
> Regarding the comparison with tracking methods, we would like to clarify two points:
> - As mentioned in L212-L213, we report the MOTA metric on the bounding box level, e.g., [6], as our goal is person association. The numbers suggested in the review are on the MOTA of body joints [1], which considers each joint independently. Thus, they are not comparable to the ones reported in our Tables.
> - To ensure a fair comparison, we used a series of state-of-the-art tracking methods (Tracktor and Trackformer are among the top performing methods on the MOT benchmarks, while AlphaPose and FlowPose are among the top performing on the PoseTrack benchmarks) and we performed a consistent evaluation. For a more broad comparison, we reported results in a variety of scenarios, including sports videos (PoseTrack), movies (AVA), and sequences that are popular for 3D reconstruction (MuPoTS). The suggested works of "KeyTrack" from Snower et al. and "Combining detection and tracking for human pose estimation in videos" from Wang et al. unfortunately have not open sourced their code, so we could not provide a direct comparison.
>
> **Evaluation - 3D metrics:**
> We clarify that our focus is on the identity association problem and how employing the recovered 3D information can actually improve and help the identity assignment. This means that the 3D pose predictor we use could also be off-the-shelf, since our goal is not about improving the 3D pose regressors.
>
> **Maximum number of people:**
> Our method does not impose a fundamental upper limit to the number of people we can track. During inference, the transformer can handle a varying number of inputs, and for the settings we experimented (e.g., PoseTrack, with maximum 50 identities and a few hundreds of frames per video), we did not observe particular limitations in terms of the performance, memory, computation, etc. Similarly, Tracktor and Trackformer do not have an explicit upper limit to the number of people that can be tracked.
>
> **UV map:**
> The UV map is estimated by regressing an appearance flow field with the same size as the texture map. In the figures, we visualize the raw output, where texture also bleeds in the other areas. However, this is not important when we apply the texture on the mesh (e.g., in Figure 4), since the invalid texels will simply be ignored. Certainly, for graphics tasks, UV prediction may be improved, but already our experiments indicate that appearances via the UV representation is useful for tracking.

---

### Official Review · Reviewer_5QdY · 2021-07-16

**Rating:** 6
**Confidence:** 4

**Summary:**

This paper presents a method for tracking multiple people in monocular videos using 3D information (3D appearance, 3D pose, and 3D location) by extending the HMA model. The paper shows the effectiveness of 3D information especially the 3D appearance in some benchmark datasets. The proposed method also shows better performance than the other state-of-the-art algorithms.
The main contribution of the paper is the use of 3D representation in improving the people tracking performance in videos of monocular cameras. The work is well written and structured. A potential place for improvement is the simple association algorithm (Algorithm 1), which may be inadequate in some complex scenes with lots of occlusions or people wearing similar clothes.

**Ethical Concerns:**

None that I'm aware of.

**Limitations And Societal Impact:**

The author adequately addressed the limitations and potential negative societal impact of their work. d

**Main Review:**

Summary:
This paper presents a method for tracking people in monocular videos using 3D representations. The main idea is to extend the popular Human Mesh and Appearance (HMA) architecture to include appearance information of people on their 3D geometry representations (3D pose and location).
This extended HMA is named HMAR (R stands for “Recovery”). HMAR model recovers the appearance of the person in the form of the texture map, which is then transformed to a compact embedding using an auto-encoder. In addition, 3D pose embedding and the 3D location of the key points are encoded in the HMAR model. Furthermore, a transformer is used to create a encode global information across space and time (video frames). Finally, the paper uses a simple association scheme to associate the 3D representations across frames using the Hungarian algorithm.
The authors conducted experiments to test two aspects of the proposed approach: 1) 3D representations vs 2D representations; 2) the effectiveness of the three components in the 3D representations. The results show that using 3D representations performs consistently better than their counter 2D representations. The ablation study shows that 3D appearance is the most important component. In addition, the proposed approach is compared with other state-of-the-art tracking methods (i.e., Trackformer, Tracktor, Alphapose, and PoseFlow) and shows better performance, especially in the IDs score.

Originality:
I agree with the authors that 3D representation is more effective than 2D representations in solving the multiple people tracking problem, especially in scenes with lots of occlusions. There exists some work that uses 3D representations to tracking multiple people in videos shoot by monocular cameras. For example, Brau [1] uses 3D to represent people and simultaneously estimate the camera parameters while inferring the 3D locations of people in 3D in monocular videos. Therefore, I think the author is not the first one that uses the 3D representation to solve tracking problems with monocular cameras. However, this paper uses a much more sophisticated 3D model than the one used in [1] and generates more 3D information of the tracked person (e.g., the key points of the pose).

Clarity:
This paper is well-written and structured.

Suggestions:
In the result videos, when people get occulted, their tracks disappeared (probably because there are no bounding boxes detected). If the same person appears shortly, they will be identified as the same person by the association algorithm. The missing part could be interpolated for the missing detection part. The advantage of using 3D representation is that they can naturally explain the 2D occlusion phenomena.

**Time Spent Reviewing:**

6 hours

---

> ### Author Response · Authors · 2021-08-10
> **Reply to Reviewer1**
>
> We thank the reviewer for the valuable comments and time.
>
> **Simple association scheme:**
> We use a simple identity association scheme in order to focus on the representation aspect and investigate the effectiveness of 3D representations for tracking. Our approach could incorporate a more sophisticated identity association method, however, we believe that the simplicity of this step actually further validates our main argument about using 3D representations for tracking - we have indeed captured very detailed, easily separable and expressive representations for each person which simplify the final step of resolving associations (Figure 1).
>
> **Previous work on tracking from monocular input using 3D information:**
> Thanks for the reference of Brau et al (the full citation for Brau et al is not provided, but we assume the reviewer refers to “Bayesian 3D tracking from monocular video” from ICCV 2013), we will cite this work in the final manuscript! As the reviewer has noted, the main difference of this work as well as most prior methods is that we leverage much more expressive and fine-grained 3D information, including detailed representations for appearance and 3D joint locations, which help to improve the overall tracking performance. We will update the wording on this in the final version.
>
> **Interpolation of the missing parts:**
> Thank you for this observation! Indeed, this is yet another advantage of using a 3D representation - we can easily interpolate the 3D location for the occluded frames. We will update the final video with examples of this illustration.

---

> > ### Comment · Reviewer_5QdY · 2021-08-23
> > **Rebuttal addressed my concerns**
> >
> > I thank the authors for their rebuttal. I think the authors did a great job in addressing my comments and concerns.

---

### Author Response · Authors · 2021-08-10
**General response to reviewers.**

We would like to thank the reviewers for their time and valuable feedback. Overall, the paper has received positive reviews, and the reviewers have appreciated the proposed idea of using 3D representations for identity association and tracking of humans: ("The paper shows the effectiveness of 3D information" (**R1**), "introducing 3D reasoning into the tracking pipeline is a very promising research line" (**R2**)), and they have praised the performance/experimental results ("improved performance w.r.t. state-of-the-art" (**R2**), "the method outperforms several recent baselines" (**R3**), "Ablation studies are quite thorough" (**R3**), "state-of-the-art results in quantitative and qualitative aspects" (**R4**)) as well as the presentation of the paper ("work is well written and structured" (**R1**), "The paper is generally clear and well-written" (**R3**), "manuscript is well-written" (**R4**)). Questions mostly regard the details of experiments. In the individual responses below, we provide replies and clarifications for the questions raised by the reviewers. All the updates, clarifications, discussions presented here, will also be incorporated in the final version.

---

### Decision · Program_Chairs · 2021-09-27

**Decision:**

Accept (Poster)

**Comment:**

This paper is somewhere in between traditional tracking, person re-identification and 3D human modeling. As claimed by the authors, the use of 3D representations in a tracking context has rarely been done before (mostly due to computational constraints). Since the method requires high-res input (or at least large humans), it may not be suited for any tracking application, which makes comparison against some state-of-the-art methods harder. The paper is accepted, but authors are expected to clearly mention and discuss limitations of the proposed method (e.g. include computational costs mentioned in rebuttal) and why some standard tracking benchmarks like the MOTchallenge were not considered.